# Silent Killer:
# A Stealthy, Clean-Label, Black-Box Backdoor Attack

## Abstract

Backdoor poisoning attacks pose a well-known risk to neural networks. However, most studies have focused on lenient threat models. We introduce Silent Killer, a novel attack that operates in clean-label, black-box settings, uses a stealthy poison and trigger and outperforms existing methods. We investigate the use of universal adversarial perturbations as triggers in clean-label attacks, following the success of such approaches under poison-label settings. We analyze the success of a naive adaptation and find that gradient alignment for crafting the poison is required to ensure high success rates. We conduct thorough experiments on MNIST, CIFAR10, and a reduced version of ImageNet and achieve state-of-the-art results.

## 1 Introduction

Backdoor poisoning attacks pose a well-known risk to neural networks (Goldblum et al., 2022; Li et al., 2022; Cinà et al., 2022). Creating undesired behaviors within deep learning models, by only modifying or adding poisoned samples into the victim's training set is especially appealing to attackers. Many datasets for training machine learning models are scraped from the internet (Radford et al., 2019; Deng et al., 2009; Schuhmann et al., 2022), thus an attacker can take advantage and upload poisoned samples and the victim will use them for training unaware that the dataset is poisoned.

Poisoning-based attacks vary in the assumed threat model, e.g. what the attacker knows about the model training. The leading categorization of this is: *White-box* – complete knowledge of the model's architecture and parameters. *Gray-box* – knowledge of the victim's architecture but not the weights, and *Black-box* – no knowledge of the attacked model architecture. These range from easiest to exploit, due to more information, to hardest. Nevertheless, in real-world scenarios, we most often face black-box settings as we don't have knowledge or control of the attacker model.

One can also categorize backdoor attacks to poison-label and clean-label attacks. In poison-label attacks, the attacker is allowed to modify the labels of the poisoned samples, whereas in clean-label settings the attacker cannot change the label, and typically cannot modify the training samples too much (under a specific norm). Clean-label attacks represent a setting where samples are curated and then manually labeled, as in standard supervised methods. Therefore, under clean-label settings, the poisoned samples must be unsuspicious and labeled as they would be without the poison. While clean-label attacks are more challenging, the above reasons make them desirable for real-world use.

Another desired quality of backdoor attacks is the stealthiness of the trigger at inference time. While early methods used a colorful patch (Turner et al., 2019; Zhao et al., 2020; Souri et al., 2022; Saha et al., 2020), many newer methods use subtle additive perturbations as triggers (Zeng et al., 2022; Luo et al., 2022; Barni et al., 2019; Liu et al., 2020; Ning et al., 2021). Zhang et al. (2021) introduce TUAP, in which they show that a universal adversarial perturbation (UAP) (Moosavi-Dezfooli et al., 2017) can make an effective trigger in poison label settings. They add a UAP to samples from the source label, change their labels to the target, and show that models trained on top of this poisoned dataset suffer high attack success rates. While this attack creates a stealthy trigger successfully in black-box settings, it only applies in poison-label settings.

In this study, we show that applying TUAP under clean-label settings has limited success. Moreover, the success varies greatly making it hard to predict if the attack will succeed. We hypothesize that in contrast to poison-label TUAP, where the only feature that can be used to succeed in the classification task is the trigger (the "adversarial features"), the success of *Clean-Label-TUAP* (CL-TUAP) depends on the ease of learning the original, non-poison features ("clean features"). If the model easily finds the clean features and therefore achieves low loss easily, or if these clean features are harder to detect and therefore the model will "prefer" to use the adversarial features to classify the samples. In the latter case, the model will connect between the UAP and the target class, and the attack will succeed. Unfortunately, It is impossible to assure that the model will indeed focus on the UAP to perform the classification and not on the clean features.

Following this hypothesis, we propose to explicitly optimize the poison samples such that the model will focus on the UAP to perform the prediction. To do that, we use gradient alignment, a method that was proposed recently for data poisoning (Souri et al., 2022). We show that by using gradient alignment on top of CL-TUAP, we improve the attack success rate (ASR) of the backdoor attack from 78.72% to 97.03% on MNIST. Furthermore, we show that this method improves the attack success on CIFAR10 on a range of architectures in gray-box and black-box settings, and achieves SOTA results under the challenging settings of clean-label, black-box attack. We demonstrate the scalability of the attack on a reduced version of ImageNet. An illustration of the attack steps is and samples of this attack are shown in Figure 1 and Figure 2 respectively. We make the code publicly available [1].

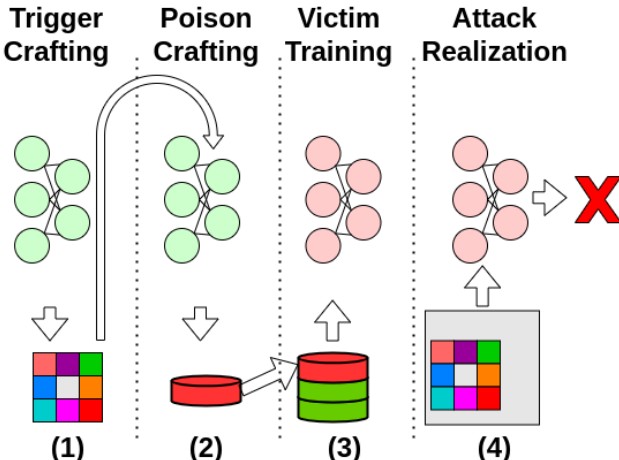

Figure 1: Illustration of the steps of the Silent Killer attack. First (1), the attacker crafts a trigger, which can be a colorful patch or a stealthy adversarial perturbation, and (2) uses it to craft the poison perturbation. Then (3), the poison is added to a small portion ($\sim 1\%$) of the training set. After the training, (4) the attacker can activate the malicious behavior using the secret trigger.

## 2 Related Work

Data-poisoning-based backdoor attacks were first introduced by Gu et al. (2017). They showed that adding a trigger to images and relabeling them as the target class can cause a model optimized on this data to predict the target class whenever the trigger is present in a sample. Subsequent works made the attack stealthier and more effective (Liao et al., 2018; Nguyen & Tran, 2021). However, most don't work under clean-label setups.

### 2.1 Clean-Label Attacks

Clean-label attacks perform data poisoning without changing the labels of the poisoned samples. To make the model focus on the trigger and not on the clean features, one approach involves "blurring" features of target

---

[1]anonymous code in supplementary

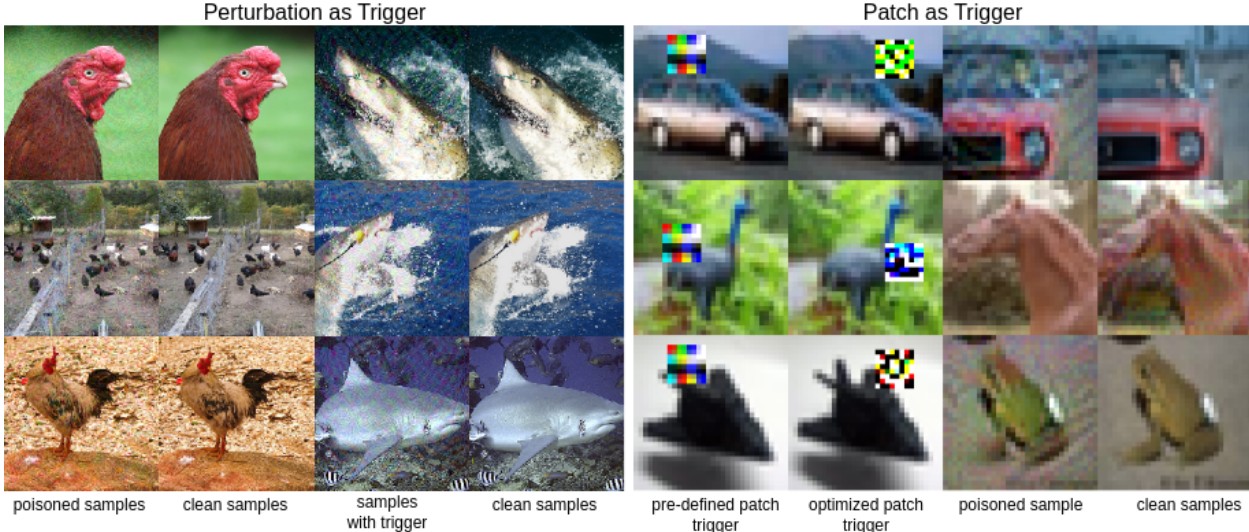

Figure 2: Left: Silent Killer attack stealthily transforming ImageNet sharks into cocks prediction. Right: Our patch attack on CIFAR10: unoptimized and optimized triggers for source classes (cars, birds, airplanes), and poisoned and clean samples for target classes (cars, horses, frogs).

class samples and then adding a trigger. Turner et al. (2019) used a GAN and an adversarial perturbation to reduce the impact of the target class features. Other works use the same idea with some changes (Zhao et al., 2020; Luo et al., 2022). A related approach is "feature collision" (Saha et al., 2020), where samples from the target class are optimized to be close to "triggered" samples in the feature space and then are used to poison the dataset. These methods have limited transferability to other architectures and struggle in black-box scenarios. Moreover, it was shown that feature collision is not effective when the victim's model is trained from scratch (not just finetuned) (Saha et al., 2020). Ning et al. (2021) also manipulates the poison samples in the feature space, and uses a UNet to produce a perturbation with amplified trigger features. However, the blending of 50% with images caused the perturbation to be bold and is a significant drawback.

An alternative approach, such as gradient alignment in the Sleeper Agent (SA) attack by Souri et al. (2022), focuses on optimizing poisons to minimize both victim and attacker losses simultaneously (achieving low errors on both test set and triggered samples). While effective in black-box settings, this approach involves multiple surrogate model retrains and requires an ensemble of surrogate models, resulting in higher crafting costs. Furthermore, the use of a visible, colorful trigger lacks stealth. Our work seeks to enhance these aspects by using optimized, stealthy triggers.

## 2.2 UAP as a Trigger

UAPs have been studied as effective inference-time attacks in various domains (Moosavi-Dezfooli et al., 2017; Maimon & Rokach, 2022). Zhang et al. (2021) introduced TUAP, demonstrating that UAPs can serve as effective triggers in backdoor attacks. They performed a UAP attack on a surrogate model, crafting a perturbation that changes predictions from the source class to the target class. By adding this perturbation to samples from the source class and altering their labels, they embedded the backdoor in the victim model. At inference time, adding the UAP to new source class samples resulted in classification as the target class, achieving high ASR without affecting clean samples' accuracy.

While TUAP utilized UAPs for poison-label attacks, Zeng et al. (2022) applied a similar concept in clean-label settings. They show that by optimizing a perturbation on samples from the target class to reduce the loss on these samples further, the crafted perturbation can be used as an effective trigger under clean-label settings. However, at inference, they had to significantly amplify the trigger ($l_\infty$ norm of 48/255 instead of 16/255). While the poison is stealthy, the trigger is relatively visible. Zhao et al. (2020) implemented

---

**Algorithm 1** Poison Crafting

---

**Input**: $F_s$, $\mathcal{D}_s$, $\mathcal{D}_t$, $\delta_t$, $t$
**Parameter**: $R$, $\alpha$, $\epsilon$

1:  Initialize $\delta_p \leftarrow \{\delta_p^{(n)} = \delta_t\}_{n=1}^N$
2:  $\mathcal{L}_a = \frac{1}{M} \sum_{x_m \in \mathcal{D}_s} \mathcal{L}(F_s(x_m + \delta_t), t)$
3:  **for** $r = 1, 2, ..., R$ optimizations steps **do**
4:      $\mathcal{L}_v = \frac{1}{N} \sum_{x_n \in \mathcal{D}_p} \mathcal{L}(F_s(x_n + \delta_p^{(n)}), y_n)$
5:      $\mathcal{A} = 1 - \frac{\nabla_{\theta_s} \mathcal{L}_v \cdot \nabla_{\theta_s} \mathcal{L}_a}{||\nabla_{\theta_s} \mathcal{L}_v|| \cdot ||\nabla_{\theta_s} \mathcal{L}_a||}$
6:      $\delta_p \leftarrow CLIP(\delta_p - \alpha \cdot \frac{\partial \mathcal{A}}{\partial \delta_p}, -\epsilon, \epsilon)$
7:  **return** $\delta^p$

---

a version of Turner et al. (2019) attack with a UAP as a small colorful patch, applied to video and image classification models. However, this method lacks stealthiness in both training and inference and does not perform well in black-box settings (Souri et al., 2022). In summary, except our method, no method performs a backdoor attack under clean-label and black-box settings, with stealthy poison and trigger.

# 3 Method

## 3.1 Attacker Goal and Threat Model

The attacker aims to poison the training set, such that after a victim trains a model $F$ on this data, it will predict the target class whenever a trigger is added to a sample from the source class. The performance of the model on clean samples must be close to the performance when trained on clean data. The number of samples that the attacker can poison is limited ($\sim 1\%$) and the perturbation that can be added is bounded ($l_\infty \leq 16/255$). These constraints are similar to other backdoor attacks (Saha et al., 2020; Souri et al., 2022). The poisoning has to be clean-label. We conduct experiments under gray-box and black-box scenarios. We assume that the attacker has access to the victim training set, which is a standard assumption in previous works (Souri et al., 2022; Saha et al., 2020). Finally, the attacker can embed a stealthy trigger into a sample and feed it to the model at inference time. The attack is successful if the model predicts this sample as the target class specified by the attacker, and the attack success rate (ASR) is the portion of the samples from the source class in the test set that are classified as the target label by the model.

## 3.2 Our Approach

Our attack consists of two stages: trigger crafting and poison crafting. First, we perform a targeted UAP attack (Moosavi-Dezfooli et al., 2017), using BIM (Kurakin et al., 2018) as our optimization method, the source and target classes of the UAP optimization are the same as the source and target classes of our backdoor attack. The perturbation is crafted on a clean surrogate model $F_s$ and used as the trigger.

The second stage consists of gradient alignment to poison samples, as described in Alg 1. We aim to align the victim's loss gradients, $\nabla_\theta \mathcal{L}(\mathcal{D}_p)$, with those of the attacker's loss $\nabla_\theta \mathcal{L}(x_i + \delta_t, t)$, where $\mathcal{D}_p$ is the poisoned dataset, and $x_i, \delta_t, t$ are a sample from the source class, the trigger, and the target class respectively. Intuitively, the attacker aims that training the model on the poisoned dataset $\mathcal{D}_p$ will implicitly optimize the attacker loss. If the gradients of both the victim's loss and the attacker's loss are aligned, then model training will also optimize the attacker's loss. We follow Souri et al. (2022) and perform gradient alignment by optimizing: $\mathcal{A}(p) = 1 - \frac{\nabla_\theta \mathcal{L}(\mathcal{D}_p(p)) \cdot \nabla_\theta \mathcal{L}(\mathcal{D}_t)}{||\nabla_\theta \mathcal{L}(\mathcal{D}_p(p))|| \cdot ||\nabla_\theta \mathcal{L}(\mathcal{D}_t)||}$ where $\mathcal{L}(\mathcal{D}_t) = \frac{1}{M} \sum_{m=1}^M CE(F(x_m + \delta_t), t)$ is the attacker loss, i.e. the loss of samples from the source class with the trigger added to them, and $\mathcal{L}(\mathcal{D}_p) = \frac{1}{N} \sum_{n=1}^N CE(F(x_n + \delta_p^{(n)}), y_n)$ is the victim's loss, which the victim optimizes directly. $CE$ is the cross-entropy loss, $N$ is the number of poisoned samples, and $M$ is the number of the samples from the source class used to estimate the target gradients. We follow Souri et al. (2022) and choose the samples with the largest gradients norm to poison. SGD is used for poison optimization: $p = p - \nabla_p \mathcal{A}(p)$

Table 1: Results of clean label attacks with and without gradient alignment, SK and CL-TUAP respectively. As one can see, gradient alignment can dramatically boost the ASR.

|  | ResNet-18 | | VGG11 | | MobileNetV2 | | CNN (MNIST) | |
|---|---|---|---|---|---|---|---|---|
|  | ASR | Clean Acc | ASR | Clean Acc | ASR | Clean Acc | ASR | Clean Acc |
| CL-TAUP | 82.43 | 83.78 | 96.3 | 86.18 | 94.11 | 82.32 | 78.72 | 98.08 |
| SK | **90.65** | 83.69 | **98.51** | 85.87 | **98.52** | 82.22 | **97.03** | 98.08 |

## 4 Results

TUAP (Zhang et al., 2021) shows very good results under poison-label settings, achieving 96%-100% ASR on CIFAR10. We wish to evaluate it under the clean-label setup, thus, we implemented the clean-label version of it, *CL-TUAP*. We first experimented on MNIST using a simple CNN with two convolutional layers (32 and 64 filters) and one output layer. We first train a surrogate model, and then craft the trigger using this model. The trigger perturbation $l_\infty$ norm is bounded by 64/255. We add the perturbation to 600 samples (1% of the dataset), and train a new model with the poisoned dataset. A similar experiment was done on CIFAR10 using different architectures: ResNet18, MobileNet-V2, and VGG11. Here, we used a perturbation of 16/255 and poisoned 500 samples (1% of the data) as in other works (Souri et al., 2022; Saha et al., 2020). We evaluated all of the 90 source-target pairs of MNIST, and chose randomly 24 pairs of CIFAR10.

**Gradient Alignment**   As shown in Table 1, especially in the MNIST experiment, the results of clean-label TUAP are not perfect. Specific source-target pairs had a significant drop in ASR, for example (source, class) of (3, 1) ASR was 24.65% and (0, 4) was 26.53%. In real-world settings, this gap may be substantial, hence we seek a way to improve this. We hypothesized intuitively that in the failure cases, the clean features are easy to find and therefore the model doesn't consider the adversarial features of the trigger. To address this issue, in Silent Killer we use gradient alignment to encourage the model to optimize the attacker loss. Table 1 clearly shows the improved performance across all models and datasets without noticeable harm in clean accuracy. The improvement on MNIST is 18.31 points, and 4.95 on CIFAR10 on average.

**Optimized Trigger**   To investigate the benefit of the UAP to the attack, we compare our results to SA (Souri et al., 2022), which performs gradient alignment with a colorful unoptimized patch that is used as a trigger. For a fair comparison, we perform a UAP to a patch with the same size ($8 \times 8$ pixels). We perform poison crafting with both the optimized and non-optimized patches using the same checkpoints of the surrogate model. Some samples can be found in Figure 2. As shown in table 2, the optimization of the trigger has a significant impact on the attack success rate, our optimized patch achieves on average 45.76 points more than the same attack without optimization, i.e. SA – (base). SA (base) is the vanilla version of gradient alignment which we also use, and SA adds surrogate model retraining and an ensemble of models for poison crafting. Even without using these tricks, which clearly improved their performance, we outperformed SA.

Note that the results of the patch version on MobileNet-V2 are significantly lower than those of the other architectures. We noticed that the success of gradient-alignment-based attacks is highly correlated with the success of the UAP crafting, which performs poorly on MobileNetV2. Therefore we estimate that better UAP methods (Madry et al., 2017; Carlini & Wagner, 2017) may yield more effective triggers and improve ASR.

To evaluate a more realistic scenario of data poisoning, when the images have a larger size, we evaluate our method on ImageNet. Due to computational limits, we used only the 10 first classes of ImageNet to train and evaluate the results. We chose one source-target pair (cock-shark), crafted poison to 185 samples from the target class, and trained a ResNet18 model. Some samples are shown in figure 2. We found that the ASR was 76% for this preliminary experiment, which indicates that the danger is also present in larger, higher resolution, real-world datasets.

Table 2: ASR comparison of Silent Killer against leading clean-label attacks in gray-box CIFAR-10 setups with 1% poisoned samples. Various trigger types are considered, including patches and perturbations ($l_\infty < 16/255$ bounded). Results for the first three methods are from Souri et al. (2022). SA (base) and Narcissus* represent our implementation of these methods, SA (base) without retraining and ensemble use, and Narcissus* with our trigger constraint ($l_\infty < 16/255$), for a fair comparison.

| Trigger Type | Architecture | ResNet18 | MobileNet-V2 | VGG11 |
|---|---|---|---|---|
| Patch | Hidden Trigger Backdoor (Saha et al., 2020) | 3.50 | 3.76 | 5.02 |
| | Clean-Label Backdoor (Turner et al., 2019) | 2.78 | 3.50 | 4.70 |
| | SA (Souri et al., 2022) | 78.84 | 75.96 | 86.60 |
| | SA (base) | 57.42 | 15.00 | 25.25 |
| | Silent Killer (patch) (ours) | **95.92** | 41.68 | 97.35 |
| Perturbation | Narcissus* (Zeng et al., 2022) | 59.06 | 85.00 | 77.56 |
| | Silent Killer (perturbation) (ours) | 90.65 | **98.51** | **98.52** |

Table 3: Mean ASR for black-box attacks, with each column representing the target architecture. The results represent the mean ASR across 24 source-target pairs. We compare our method to SA (Souri et al., 2022) when they use a single surrogate model (ResNet18) as we do, and when crafting on an ensemble (ResNet18, ResNet34, VGG11, MobileNetV2, excluding the victim). Notably, our approach outperforms SA even when they use an ensemble.

| | Surrogate \Target | ResNet18 | VGG11 | MobileNetV2 |
|---|---|---|---|---|
| SA (Souri et al., 2022) | ResNet18 | - | 31.96 | 29.10 |
| | Ensemble | 63.11 | 55.28 | 42.40 |
| Silent Killer (ours) | ResNet18 | - | 81.60 | 90.92 |
| | VGG11 | 76.97 | - | 95.56 |
| | MobileNetV2 | 32.44 | 77.53 | - |

**Black-Box** To evaluate Silent Killer's black-box effectiveness, we used ResNet18, VGG11, and MobileNetV2 models in all surrogate and target combinations (Table 3). Our black-box attack consistently outperforms SA. For example, we achieved a substantial 61.82 ASR improvement for MobileNet and 49.64 ASR points for VGG11 when the surrogate model was ResNet18. Notably, our method achieves these results without the need for the computationally costly retraining procedures used by SA. This suggests our attack's superiority in black-box settings and the potential for even better performance by adopting similar techniques.

**Sensitivity Analysis** In real-world scenarios, discreetly adding many poisoned samples to the victim's training data is challenging. We examined how the ASR varies with the number of poisoned samples ($N$), testing with $N = \{50, 100, 200, 300, 400\}$ on three models. In Figure 3, we demonstrate our method's efficiency, achieving success with as few as 50 samples, only 0.1% of the training set's size. This highlights our attack's effectiveness with a limited number of training samples.

We also examined the upper limit of perturbation norms for the trigger and poison, using $\epsilon \in \{4/255, 8/255, 12/255, 16/255\}$. As expected, we observed a tradeoff between attack stealthiness (determined by $\epsilon$) and performance. Results and perturbation examples are in Figure 3.

**Defences** We evaluate the effectiveness of three different types of defenses against our method, based on three different concepts: gradient shaping, data filtering, and data augmentation. We chose methods that were found to be effective at evading gradient-alignment-based data poisoning (Souri et al., 2022). The results in table 4 show that all these methods did not significantly deteriorate our attack's ASR.

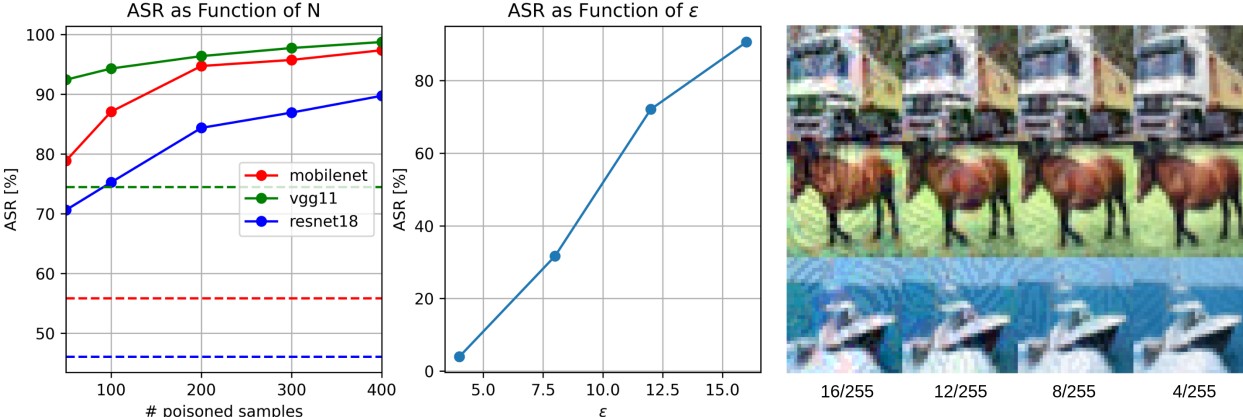

Figure 3: Left: Sensitivity analysis of the number of poisoned samples $N$. The dotted lines represent the results of using the optimized trigger without poisoning. Even with a small amount of data (e.g. 100 samples, 0.2% of the dataset), reasonable results can be obtained. Center: ASR as a function of the poison and trigger perturbation magnitude ($l_\infty$). Training was of ResNet18 on CIFAR-10. The units on the x-axis are in the range (0, 255). The larger the norm, the better the performance. Right: Samples of images with triggers with different $l_\infty$ norm bounds.

Table 4: ASR and clean accuracy after applying defenses. Clean accuracy without defense is 82.64%. As we can see, the success of the attack does not significantly deteriorate after using these defenses.

| Method | ASR [%] | Accuracy [%] |
|---|---|---|
| Activation Clustering (Chen et al., 2018) | 71.28 | 73.13 |
| DPSGD (Hong et al., 2020) | 91.16 | 84.36 |
| MixUp (Borgnia et al., 2021; Zhang et al., 2017) | 94.24 | 81.39 |

## 5   Conclusion

We present a novel data poisoning attack that uses UAP trigger optimization and gradient alignment for poison crafting. Our attack is stealthy and highly effective under realistic settings, i.e. clean-label, black-box, and stealthy poison and trigger. Despite its success, poison-label methods still outperform the SOTA clean-label methods. It is interesting to investigate if better clean-label methods can match this gap. In future research, we could explore using more sophisticated methods for UAPs as triggers. We could also jointly optimize both the poison and the trigger to further boost performance. It is critical to continue developing robust defenses against data poisoning.

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

# A   Appendix

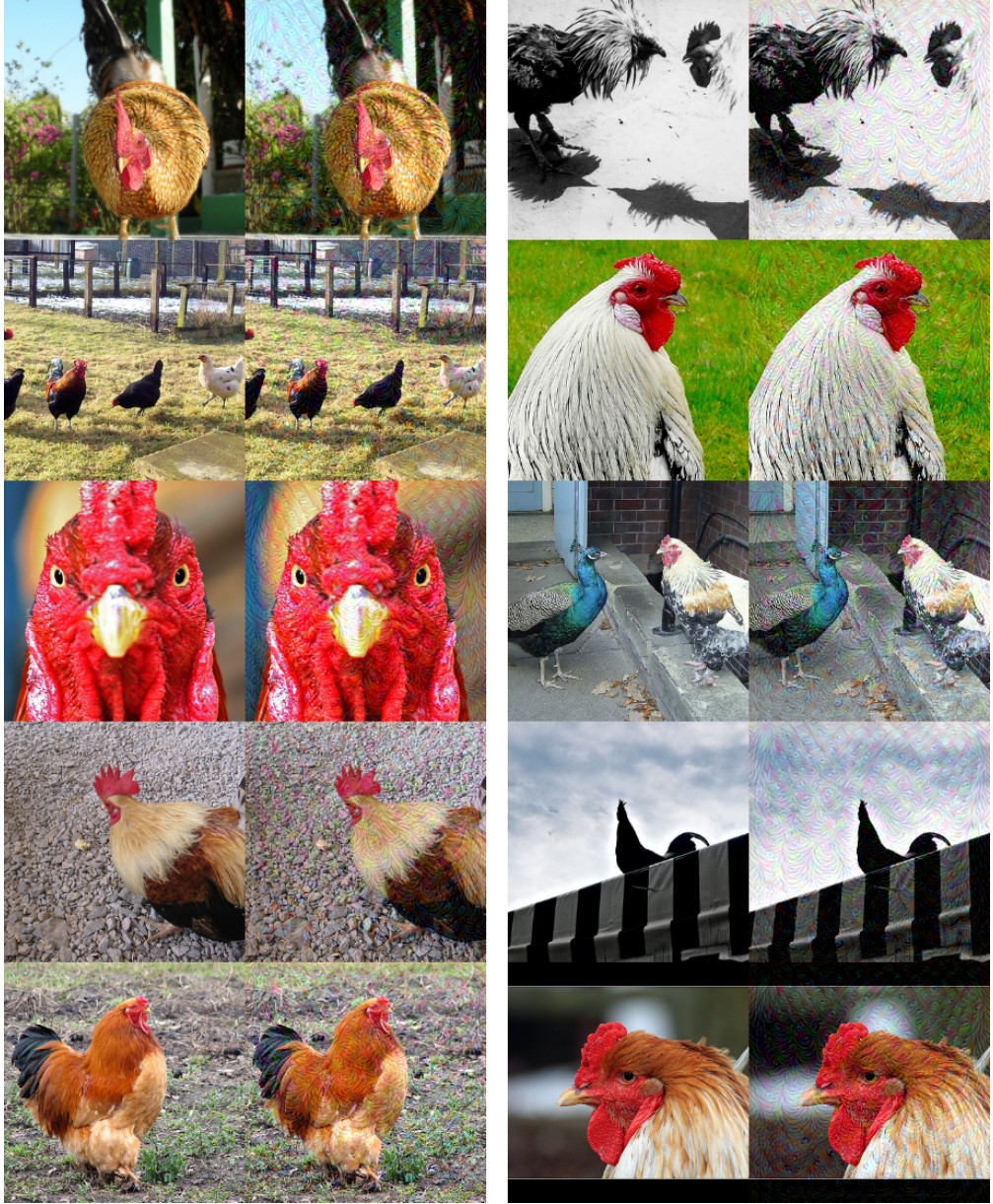

Figure 4: Samples of images from ImageNet with and without the trigger.

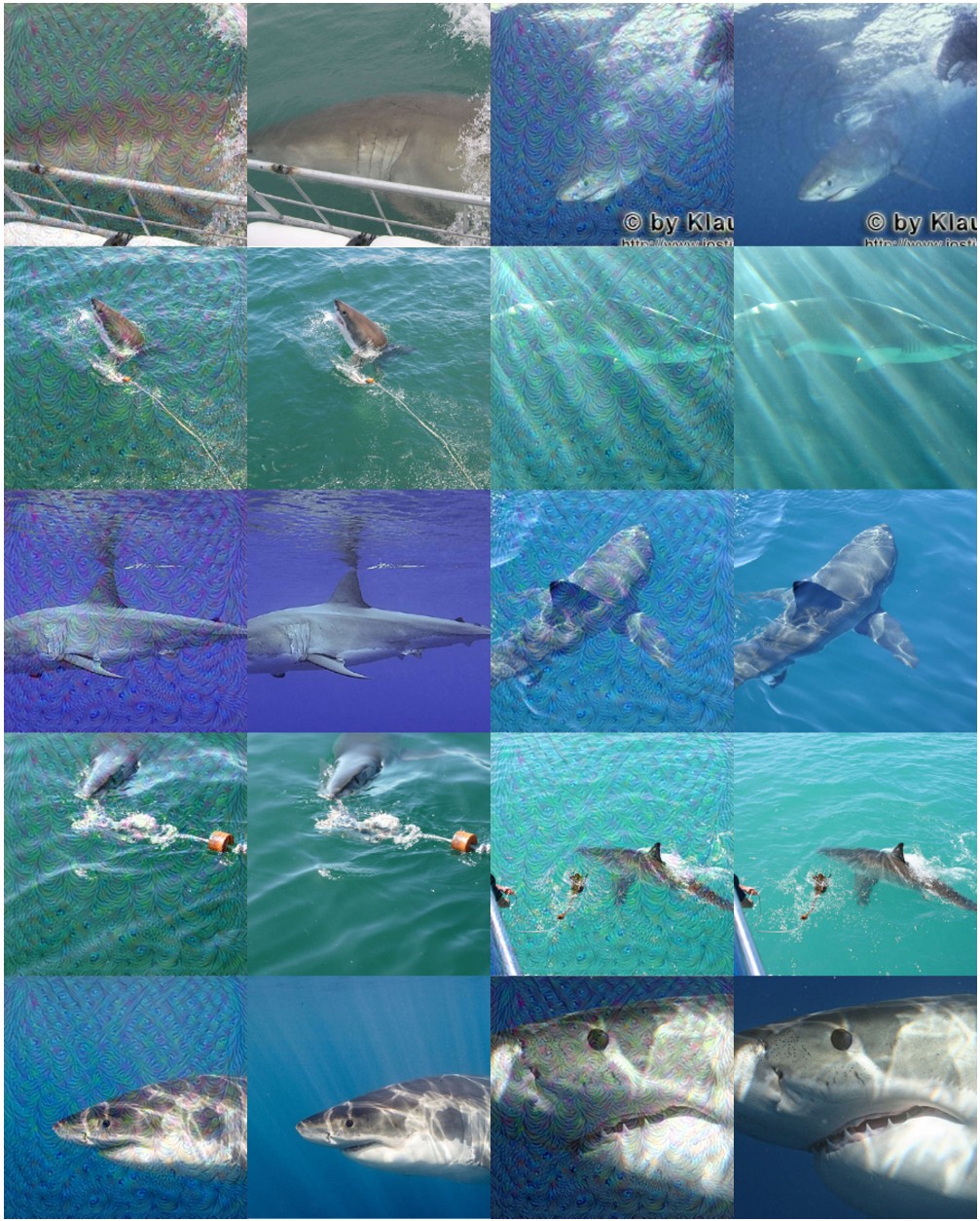

Figure 5: Samples of images from ImageNet with and without the poison.

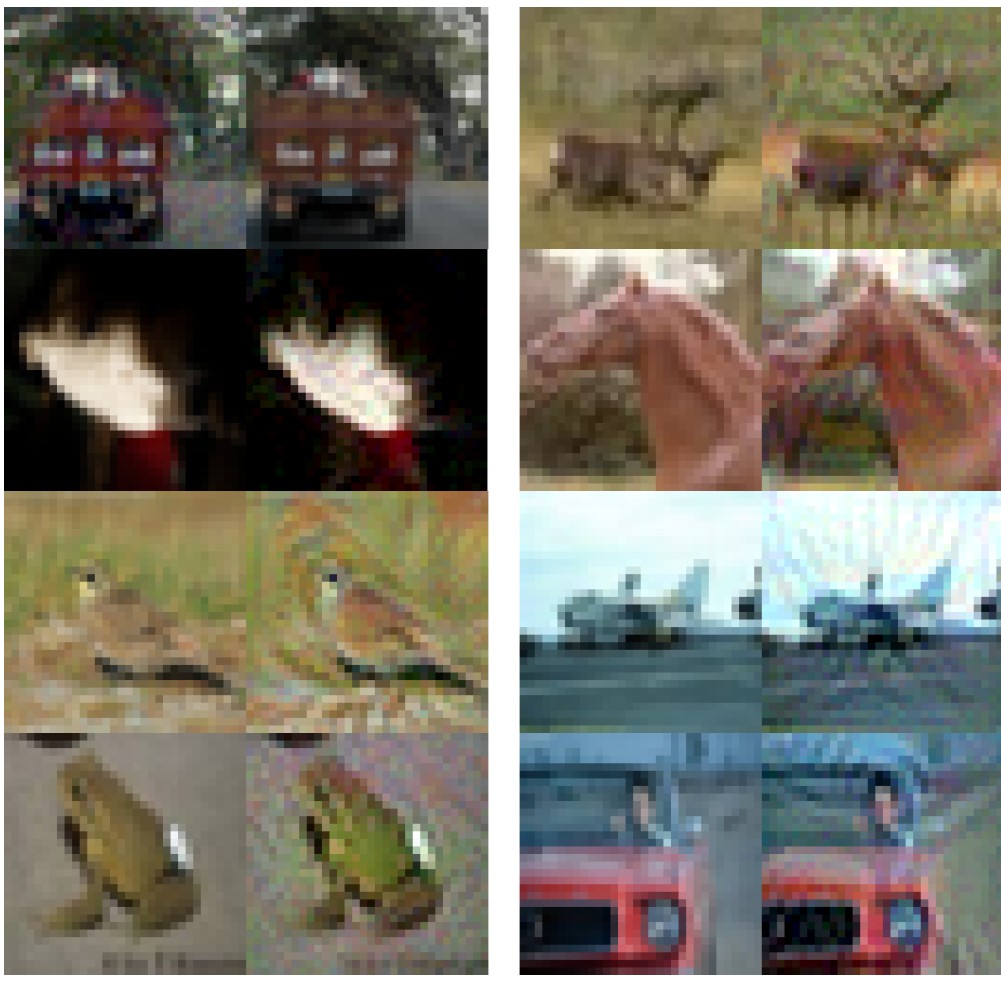

Figure 6: Samples of images from CIFAR10 with and without the poison, when the trigger is an optimized colorful patch (not in this figure).

