# OpenReview forum: "Silent Killer: A Stealthy, Clean-Label, Black-Box Backdoor Attack"
_TMLR — Withdrawn by Authors_

### Review · Reviewer_fP9w · 2024-03-14

**Summary Of Contributions:**

This paper introduces a novel backdoor attack method named Silent Killer, utilizing Universal Adversarial Perturbations (UAP) and gradient matching techniques to create covert and efficient clean-label backdoor attacks. The author observes that gradient matching enhances the success rate of backdoor attacks. Experimental results on three image datasets (MNIST, CIFAR10, and a subset of ImageNet) confirm the effectiveness of the proposed approach.

**Audience:**

Yes

**Broader Impact Concerns:**

No concerns on the ethical implications of the work are identified.

**Claims And Evidence:**

No

**Requested Changes:**

-	Unclear method flow in section 3. For instance, what does trigger crafting entail in the first phase? Why is targeted UAP prioritized? Additionally, the statement regarding "the source and target classes of the UAP optimization are the same as the source and target classes of our backdoor attack" is ambiguous. What is the relationship between the optimization target of UAP and the backdoor target? The authors are advised to restructure the method flow.

-	Lack of prerequisite knowledge about gradient alignment. The authors should provide an introduction to gradient alignment to aid readers' understanding of the proposed attack method.

-	Unclear threat model. The authors should specify the capabilities of the attacker and applicable attack scenarios. The current threat model section lacks sufficient information on attacker capabilities and requires further refinement.

-	Lack of technical advantages. What distinguishes the proposed attack method from [1]? What constitutes the technical innovation of this work?

-	Missing details about experimental setup. The authors need to introduce experimental data and parameter settings to facilitate readers' understanding of the experimental information.

-	Lack of effective defense baseline. The authors have not considered the most advanced defense methods, such as defenses based on knowledge distillation [2] and pruning [3]. Incorporating these stronger defense methods would help demonstrate the effectiveness of the proposed approach.

[1] Sleeper Agent: Scalable Hidden Trigger Backdoors for Neural Networks Trained from Scratch. NeurIPS 2022.
[2] Neural attention distillation: Erasing backdoor triggers from deep neural networks. ICLR 2021.
[3] Reconstructive Neuron Pruning for Backdoor Defense. ICML 2023.

**Strengths And Weaknesses:**

Strengths:
- The research motivation is clear.
- Exploring for more covert and efficient backdoor attacks is an intriguing topic.

Weaknesses:
- The writing requires further improvement.
- The description in the method section lacks clarity.
- There is a lack of effective defense baselines.

---

### Review · Reviewer_5rcE · 2024-03-15

**Summary Of Contributions:**

The paper proposes "Silent Killer", a new method for nearly-imperceptible, clean-label, black-box backdoor attacks on image classifiers. The method combines the universal-adversarial-perturbation setting with gradient-alignment based attack optimization. The method is evaluated on MNIST, CIFAR10, and a reduced version of ImageNet, where good results are obtained.

**Audience:**

Yes

**Broader Impact Concerns:**

The proposed "Stealthy, Clean-Label, Black-Box Backdoor Attack" is a realistic threat model, which could cause harm in high-stake applications when being exploited. The authors also show that existing defences are ineffective. The authors should discuss the broader impact of their work accordingly.
The writing of the paper also identifies sometimes too strongly with the role of the attacker: e.g. "While clean-label attacks are more challenging, the above reasons make them desirable for real-world use." - these attacks are not "desirable" but "plausible and concerning". The authors should revise the paper in a way that makes clear that they are aware of the potential negative downstream implications of their work.

**Claims And Evidence:**

No

**Requested Changes:**

- Evaluation on larger datasets or tasks beyond image classification to show the generality of the proposed procedure
- Attacks on modern networks/training pipelines - clean accuracy of <90% is too far from  state-of-the-art performance on CIFAR10 to provide meaningful insights
 - Clarify contributions of Silent Killer over prior works such as Sleeper Agent.

**Strengths And Weaknesses:**

Strength:
 * The authors study a relevant ML security topic (backdoor attacks on image classifiers). The setting is well motivated and introduced and overall, the paper is easy to follow
 * The proposed method makes intuitively sense - however, it remains unclear to which extent it exceeds prior works like Sleeper Agent or is rather a special case of it.
 * The method is shown to improve over baselines and to remain effective against defences.
 * The effect of perturbation strength and number of poisoned samples is studied (Figure 3).

Weaknesses:
 * Experiments on MNIST, CIFAR10, and a reduced version of ImageNet are too small-scale to provide much insights into applicability of the method to modern large-scale datasets
 * Baseline performance of attacks networks is extremely low (<=86% on CIFAR10) - it remains unclear why and if the presented attacks would remain effective for stronger architectures/training protocols
 * The method section is very short - it does not make clear what the main contributions over Sleeper Agent (SA) by Souri et al. (2022) are (which seems to be very closely related). Moreover, the method "Silent Killer" is never clearly defined, that is the name "Silent Killer" is never used in the Method section.

---

### Review · Reviewer_nG8r · 2024-03-17

**Summary Of Contributions:**

This paper studies the triggers of backdoor attacks. The authors propose to use an image-size perturbation (similar to those used in adversarial examples) which is imperceptible to human eyes as triggers during inference such that the attack becomes harder to detect. The paper proposed to use UAP as triggers and perform a gradient matching algorithm.

**Audience:**

No

**Broader Impact Concerns:**

No concerns.

**Claims And Evidence:**

No

**Requested Changes:**

I think designing a clean-label attack with stealthy triggers still has its value, but the presentation is not acceptable. I would suggest the paper be rewritten entirely to satisfy the basic bar of the venue. Details are provided above.

**Strengths And Weaknesses:**

**[General quality of the paper and presentation]:** First of all, before touching on the technical details and contribution of the paper, I want to mention that this paper is surprisingly low-quality in terms of presentation with incomplete related works, super simplified main method, undefined notations, and unclear experimental settings. I had to read several references to clearly understand the necessary background and the main contribution (being a researcher in this field of study), and I was astonished to find that several parts of the paper are simply rewritings of Souri et al. with some inferior trims (which I will detail below). Based on these aspects, I am strongly against accepting the paper in its current state and I think it is not even ready as a decent workshop paper, not to mention a journal paper. I am puzzling why the authors would consider submitting the paper as it is apparently below the bar. Also, I highly doubt that the paper can be improved in a short period of time, but in case the authors want to give it a try, here are some suggestions:

- *[Claim in the introduction]*: In the fifth paragraph of the introduction, the authors said: "Unfortunately, It is impossible to assure
that the model will indeed focus on the UAP to perform the classification and not on the clean features" and this claim might not be true. Imperceptible perturbations have been shown to create shortcuts in classification problems in [1], and it is possible to study this phenomenon in the context of backdoor attacks. The paper might be more interesting if the authors could touch a bit on this aspect and come up with a better clean-label TUAP attack. \
[1] Availability Attacks Create Shortcuts, https://arxiv.org/abs/2111.00898
- *[Related works]*: Section 2.1 cited the exact same references as Souri et al. without the paragraph about other data poisoning attacks, which is actually quite important (and can be further expanded). There exist so many more backdoor attacks in the literature and the authors should not rely on the literature review (even a trimmed one) of one single reference.  Moreover, I don't understand why 2.1 and 2.2 need to be separated. They can be merged as a single paragraph named "backdoor attacks" to avoid repeated entries (e.g., Turner et al.).
- *[Method]*: The current Section 3 looks like a technical report rather than a research paper. (1) Section 3.1 lacks a basic description of the threat model. For example, how is the attack realized? Should the model be trained with clean and poisoned data? What is a clean label attack? What is gray-box and black-box? (2) Same problem with Section 3.2, what is BIM? What is $\mathcal{D}_s$ (I have to go to Souri et al. to find the definition)?  Why don't you list proper equations? If the authors prefer to simplify things, I suggest you change Section 3 to one sentence: "Our method is a straight adaptation of Souri et al. with a trigger generated by UAP". Moreover, since the method directly applies Souri et al., why do you claim that your method can enhance the crafting cost in Section 2.1?
- *[Experiments]*: It is very surprising that a different trigger would improve Souri et al. by such a big margin. Does this suggest the gradient alignment algorithm is very sensitive to the choice of the trigger pattern? And why is a **optimized** trigger so much better than a colorful unoptimized patch? The trigger is not optimized during the craft of the poisoning, why does it affect the final results?
- *[Contribution]*: It appears to me that the entire paper is an ablation study to Souri et al. in terms of the choice of the trigger. I hope the authors can convince me why we should publish this as an individual paper.

---

### Note · Authors · 2024-03-26

**Comment:**

Upon careful consideration of the feedback provided in the reviews, we have come to the conclusion that our paper requires further development and refinement. We appreciate the insightful comments and suggestions offered by the reviewers, and recognize that addressing these points adequately will necessitate more time than the allotted two-week period. Consequently, we have made the decision to withdraw our paper. We respect the time and effort invested by the reviewers in evaluating our work. We would like to thank the reviewers for the thoughtful remarks provided in the reviews. We look forward to the opportunity to resubmit an improved version of our work in the future.

**Withdrawal Confirmation:**

I have read and agree with the venue's withdrawal policy on behalf of myself and my co-authors.